# Caring for the Elderly Enhances Positive Attitudes Better Than Knowledge in Nursing Students

**DOI:** 10.3390/medicina58091201

**Published:** 2022-09-01

**Authors:** Elena Castellano-Rioja, Marta Botella-Navas, Lourdes López-Hernández, Francisco Miguel Martínez-Arnau, Pilar Pérez-Ros

**Affiliations:** 1Department of Nursing, Universidad Católica de Valencia San Vicente Mártir, 46007 Valencia, Spain; 2Department of Physiotherapy, Universitat de València, 46010—Gascó Oliag 5, 46010 Valencia, Spain; 3Frailty and Cognitive Impairment Research Group (FROG), Universitat de València, Menendez y Pelayo 19, 46010 Valencia, Spain; 4Department of Nursing, Faculty of Nursing and Podiatry, Universitat de València, 46010 Valencia, Spain

**Keywords:** ageism, nursing students, attitudes, aged, intervention

## Abstract

*Background and Objectives*: There is a growing interest in the measurement of attitudes towards older people in healthcare professionals, as there is a need to implement interventions to improve ageist attitudes. A one group pretest–posttest study was carried out to explore the change in nursing student attitudes towards the elderly during their university careers. *Materials and Method*: A total of 97 nursing students were enrolled. Attitudes were analyzed using Kogan’s Attitudes Toward Old People Scale. *Results*: The attitude was positive from the beginning of the study, with a score of 110.06 (12.92). No differences were observed after completing the subject “Care of the elderly” (106.21 (10.77)), though a significant increase was recorded after the completion of clinical placement (142.88 (12.64)), with a large effect size (η2p = 0.754). The score of the positive items was significantly increased, but not that of the negative items, as ageist attitude was not reduced. *Conclusion*: The current curricular design, that includes taking the theoretical course before clinical placement in the geriatric area, increases positive attitudes in nursing students but does not reduce ageist attitudes.

## 1. Introduction

The aging of the population is a challenge, with social, economic and, especially, health care delivery concerns. Indeed, the increase in life expectancy generates a greater and more complex demand for healthcare services [1]. Worldwide data corroborate that by 2050, the number of people over 60 years of age is expected to double, reflecting a progressively aging process [2].

Aging comprises physiological, psychological, emotional cognitive, and social changes, some of which are linked to the aging process itself without being pathological, though they increase the risk of comorbidities [3]. The older population has a higher prevalence of chronic disorders that generate a greater need for health care, mainly in the form of nursing care [4,5]. However, not all healthcare professionals see this care as a desirable field for their professional development. Several studies have analyzed the preferences of areas of work in healthcare profession students, evidencing little interest in working with the older population [6]. This being particularly so among nursing students and even among professionals, despite the current growing demand [7,8].

This low preference on the part of health professionals could be related to the stereotypical image of the older person, defined as “ageism” towards our elders [9,10,11]. The most frequent stereotypes consider the older person as uninteresting, intolerant and non-productive [10], which can lead to discriminatory practice that negatively affects the health of older adults by providing inadequate care [12].

There is a growing interest in the measurement of attitudes towards older people in healthcare professionals, as there is a need to implement interventions to improve ageist attitudes [13,14,15]. Empathetic attitudes towards older persons can be assessed by using validated scales [16]. In addition, a variety of interventions are available, seeking to improve empathetic attitudes, including those aimed at increasing knowledge about aging and the care of older people through contacts with this population group, and interventions that introduce new technologies as simulation in the classroom [17].

However, we should also be aware that several factors are involved in the attitudes of nursing students [18], and that during the university training process itself there are interventions inherent to the curriculum that could interfere with these attitudes [19,20]. Specifically, the Nursing university degree includes the subject “Care of the elderly”. This subject, together with the practicums in which students make contact with older people, should allow them to understand the changes associated with the aging process and plan appropriate care to treat or prevent health problems and integrate these changes in their daily lives [21].

Despite this, and although attitudes tend to be positive, there are mixed results regarding the improvement of attitudes towards older people among nursing students following various types of interventions [13,15,17]. Several studies have evidenced an increase in positive attitudes after an intervention, but this is not reflected in the subsequent desire to work in geriatric care [22,23].

The nursing curriculum should be designed to target ageist attitudes by promoting socialization with older people and creating more supportive learning environments in the care setting of older people [24]. It is convenient to know which interventions (taking theoretical subject and clinical placement in the geriatric setting) in the nursing curriculum can modify the attitude of nursing students in order to be able to plan the most appropriate interventions. The present study was, therefore, carried out to assess the attitude of nursing students and the influence of taking the theoretical subject “Care of the elderly” and Practicums II and III (PII and PIII), involving clinical placement in the geriatric setting (nursing home or long term care) and in hospital.

## 2. Materials and Methods

### 2.1. Design

A one group pretest–posttest study was conducted on nursing students from the Universidad Católica de Valencia San Vicente Mártir, Spain, between October 2018 and October 2020. Inclusion criteria were the following: at least a second year student, with enrollment in the adult care course at the start of the study. The exclusion criterion was failure to sign the informed consent form.

The nursing degree at the Universidad Católica de Valencia San Vicente in the 2019 academic year had approximately 720 undergraduate students in total, distributed into 180 students in each of the four courses of the degree.

Nursing studies at the Universidad Católica de Valencia San Vicente take place over four academic years (Figure 1). The curriculum distribution is approved by the Regional Ministry of Education following the recommendations of the European Higher Education Area. As in the rest of the Health Sciences degrees, 90% of the basic modules, such as anatomy, physiology, etc., are taught in the first year. Throughout the second year, 30% of basic Health Science subjects are taught and the rest corresponds to specific subjects of the nursing discipline. It is here that they study the subject “Care of the elderly”, “Care of adults” and “Care of children”. In addition, there is a practicum rotation where students complete a 330-h clinical placement in a hospital (with the aim of acquiring general skills in hospital work methodology, work equipment, facilities, medication administration, basic care, etc.). In the third year, the students complete nursing care subjects and two clinical placement rotations: one of them in long term care (330 h) and the other in a hospital (490 h). In the fourth year, students take other specific nursing care subjects (emergency and mental health care), complete an undergraduate thesis and carry out the final clinical placement rotations, one of which is in a hospital (Intensive Care Unit or Emergency service) with 660 h of work, and the other in a primary care setting with 490 h of work.

### 2.2. Sample Size

The sample size was calculated using the XLstats^®^ application, based on a previous study where the Kogan’s Attitudes Toward Old People (KAOP) score in nursing students was 103.79 (6.27). A statistical power of 80% with α = 5 and an expected loss rate of 40% were adopted. Thus, the minimum sample required for this study was found to be 34 nursing students.

### 2.3. Ethics

The study was approved by the Research Ethics Committee of the Universidad Católica de Valencia San Vicente. The data obtained were processed in compliance with Spanish legislation, specifically Ley Orgánica 3/2018, of 5 December, on personal data protection and guarantees on digital rights, and Law 14/2002, of 14 November, regulating patient autonomy, rights and obligations referred to clinical information and documentation. The study likewise complied with the ethical principles guiding research specified in the Declaration of Helsinki. All participants gave written informed consent before inclusion in the study and statistical processing of the data.

### 2.4. Data Collection

#### Instrument and Variables

The instrument selected for attitude assessment was Kogan’s Attitudes Toward Old People scale [25]. The instrument consists of 34 items, 17 of which are positive in terms of attitude towards older adults, while 17 are negative. The scale is designed as a summed Likert attitude scale on 6-point response categories that range from 1 (strongly disagree) to 6 (strongly agree). Scores on the negatively worded items had to be reversed to obtain the total score. The possible score was between 34 and 204. Higher total scores indicated a more positive attitude. A score greater than 102 is considered a positive attitude. The Kogan’s Attitudes Toward Old People scale obtained a reliability value of 0.82 [26]. The sociodemographic variables of age and sex were also collected, as well as internship and location, previous experience in contact with the elderly, and the desire to work at the end of their studies in the field of geriatrics and gerontology.

### 2.5. Procedures

Data collection began in Moment 1 (M1) prior to the start of Practicum I and the “Care of the elderly” course in the first week of October 2018 (Figure 1). The 3 groups in the second year consisted of a maximum of 60 students. Two of the researchers scheduled the date with the teachers to allow them to enter the classroom and to be granted a maximum of 30 min. A booklet containing the information sheet, the informed consent form and the variable collection sheet, together with the Kogan scale, was handed out in paper format. Of the 30 min, the first minutes were dedicated to explaining the aim of the study, as well as to signing the informed consent form, and then the Kogan scale and the rest of the variables were filled in.

Moment 2 (M2) was at the beginning of the third course in the first week of October 2019, with the students already having taken the subject “Care of the elderly”, and prior to Practicums II and III (PII and PIII), where they have contact with older adults.

Moment 3 (M3) was at the beginning of the fourth academic year in the first week of October 2020, where the participants had completed Practicums II and III. Due to the public health alarm situation in March 2020 (COVID-19 pandemic), not all students were able to perform the two practicum periods, due to the period of home confinement in Spain during mid-March 2020 and May 2020. This information was also collected and analyzed.

### 2.6. Data Analysis

Descriptive statistics were computed to characterize the sample and data distribution and to check assumptions. The variables were reported as proportions and/or means and standard deviation (SD). The Kolmogorov-Smirnov test was used to assess normality. Parametric testing (Student t-test for independent samples) was used to compare quantitative variables, while nonparametric tests (chi-squared test) were used for categorical variables. Repeated measures multivariate analysis was performed to test the differences in outcome (Kogan’s score). Multivariate analysis of variance (MANOVA) likewise tested the differences in outcome (Kogan’s score) according to preference of work and previous experiences with older people. Effect sizes (η^2^_p_) for analysis of variance (ANOVA) were also calculated, with values of 0.01 to 0.057 considered to represent a small effect; 0.058 to 0.137 a moderate effect; and 0.138 or more a large effect. In addition, the validity of the instrument was analyzed using Cronbach’s alpha.

The study data were entered in MS Excel spreadsheets, and the statistical analysis was performed using the SPSS version 23.0 statistical package (IBM Corp. Released 2010. IBM SPSS Statistics for Windows, Armonk, NY, USA).

## 3. Results

Ninety-seven students were recruited in M1, with a mean age of 22.5 (4.92) years; 20.6% (n = 20) were aged 25 years or older, and females predominated. Most of the students had previous contact with older adults in the community and did not consider working with them as their first choice, being relegated to the case of not finding work in another area (Table 1).

After analyzing the attitudes in the three moments, the mean Kogan’s score was found to be 110.06 (12.92) in M1, 106.21 (10.77) in M2 and 142.88 (12.64) in M3. A positive attitude was therefore observed at all three timepoints, since the scores were above the cut-off point set at 102. Cronbach’s alpha was 0.71, 0.70 and 0.73, respectively. Differences were found between M3 and M1 (mean difference (MD) = 32.82, 95%CI: 24.43–41.21; *p* < 0.001) and between M3 and M2 (MD = 36.68, 95%CI: 30.23–43.12; *p* < 0.001). The model was statistically significant, with great power (F = 101.24; *p* < 0.001; η2p = 0.754) (Figure 2).

After analyzing the positive and negative items separately, it was seen that the increase in the total Kogan’s score was derived from the increase in M3 in the positive items (F = 17.64; *p* < 0.001; η2p = 0.355) (Figure 3). The mean Kogan’s score was 62.39 (12.78) in M1, 62.18 (8.29) in M2 and 70.27 (9.16) in M3. In order to analyze the negative items, inversion of the score of each item was performed, so that a negative item with a high score implied no negative attitude or ageism. We observed an increase from M1 to M2 (70.06 (7.96) up to 74.75 (7.75); *p* = 0.065), indicating a decreasing tendency in ageist attitudes, but this tendency disappeared in M3 (72.13 (8.31)) (Figure 3).

The highest scoring negative items were item 15 (Most of the elderly easily make people ill/make them feel unwell) and item 23 (It is best to move to an area where there are not many elderly people in order to keep your neighborhood nice), while the lowest score corresponded to item 5 (Most of the elderly live as they wish and cannot change). Significant differences were observed in 6 of the 17 items, but in items 1, 9, 13 and 19, despite the decrease in ageist attitude after M2, an increase in ageist attitude was observed after M3 (Figure 4 and Appendix A).

The highest scoring positive items were item 18 (One of the most interesting and entertaining qualities of most elderly people is that they tell you about their past experiences) and item 22 (When you think about it, old people have the same faults as anybody else), while the lowest scores corresponded to item 32 (Most of the elderly rarely complain about the behavior of the younger generation) and item 34 (Most of the elderly need as much love and reassurance as other people). An increase in positive attitude was observed in 9 of the 17 items (Figure 4 and Appendix A).

After performing the analysis according to previous experience with older adults (F = 0.88; *p* = 0.461; η2p = 0.081), no differences were found in the three moments. A slight nonsignificant decrease after taking the specific subject and the general Practicum I, and a general significant increase after the completion of Practicum II and III were observed. Previous experience with older adults did not interfere in the attitudes after completion of the internship period in contact with older adults, increasing the attitude in all previous experiences, but, especially, in those who previously had not had contact with the elderly (Figure 5a).

Due to the COVID-19 pandemic, 20.6 (n = 7) of the subjects in M3 did not take Practicum III, so we analyzed the difference in attitude in M3 according to the place where the practicum was taken, without finding statistically significant differences (PII 144.14 (16.2) vs. PII and PIII 142.56 (11.9); *p* = 0.772).

Similarly, no differences were found after analysis of the choice of future work in the area of geriatrics (F = 1.46; *p* = 0.226; η2p = 0.086). However, a slight nonsignificant decrease was observed after taking the course and general Practicum I, and a significant overall increase was recorded after completing Practicums II and III (Figure 5b).

## 4. Discussion

There are many factors that determine the attitude of nursing students towards older adults, such as age, gender, exposure to older adults, geriatric education, the attitude of the clinical instructor, and even the level of knowledge and empathy. The need to integrate a study program in geriatric nursing into clinical curricula for nursing students in order to develop positive attitudes towards older people, and to improve their motivation to work with older people, has been evidenced in the literature [27,28]. The aim of our study was to determine the attitude of nursing students and the influence of the current curricular program after taking the “Care of the elderly” subject and Practicums I, II and III, where the students were exposed to older adults. The attitude was positive, and there was a decrease in ageist attitude after taking the course, together with an increase in positive attitude after completing the practicum, with a strong effect size.

Our data support the results found in the literature on observing an improvement in attitude towards older adults, though we note two distinct or previously non-discussed aspects. The first is that such improvement is derived from positive attitude and not from the reduction of negative attitudes or ageism. Secondly, contact with older adults seems to be essential in modifying attitudes, regardless of the factors that may interfere.

The role of increased knowledge and empathy-enhancing experiences, and their influence upon attitudes towards older adults is a much-discussed aspect of the literature [13,14,17,29]. As the number of years of career increases, greater empathy is observed [30] and, therefore, an increased positive attitude towards older adults can be expected [31]. However, the increase in knowledge does not always generate an increase in positive attitude, since it depends on the baseline situation, the level and type of content taught in the training, the teaching methodologies used, and even the attitude and predisposition of the lecturer on the subject, etc. [13,14,29]. Our results were in agreement with this statement, as we found that increased knowledge did not generate change, and it was after contact with older adults in clinical placements that positive attitude increased. Therefore, empathy may modulate the association between knowledge and attitudes toward older adults [32]. However, in order to understand these effects in depth, there is a need for randomized trials that analyze the different types of interventions separately [29].

It may be that the large effect size found in our study was due to the order of the courses, with “Care of the elderly” being taken first, followed by clinical placement with older people. Keeping-Burke et al. [33] highlighted the importance of students entering the nursing home or long term care setting after acquiring knowledge of how to care for older adults in order to be able to respond to challenging resident behaviors. Equally important is the need for students to understand the roles and contributions of all care staff in the setting, including nurses and unregulated care providers. While the residential care setting for older adults can be a challenging learning environment for students, it also offers opportunities for student professional growth and development, especially when there are clearly articulated learning outcomes and appropriate role models are available [33,34].

### 4.1. Negative and Positive Scores

There was a strong effect in the improvement of attitude in the evaluation of the positive items, with the exception of item 34 after the clinical placements. This increase could be due to the fact that contact with the elderly in the internship, on one hand, allows for better knowledge of the particularities of old age [35], contrasting the image students have of older people [36], and, on the other hand, favors the development of empathy, which has a direct impact upon attitudes towards the elderly [17,37].

The score derived from the negative or ageist items did not change throughout the study, though there are studies indicating that higher levels of empathy indicate a downward trend in positive and negative ageism in students [38]. In the detailed analysis, there was a significant increase of ageist attitudes in three of the total 17 items: item 1 (It would be better if most of the elderly lived with their coevals in the same place), item 9 (Most of the elderly tend to let their houses become untidy and unkempt) and item 19 (Most of the elderly spend too much time interfering in other people’s business (stick their noses in)). The implementation of practices in hospital and long-term care settings took place in the post-pandemic period (COVID-19). During this period the increase in the use of home care in older adults was greater due to the fact that hospital admission was avoided in order to minimize exposure in the hospital environment [39]. This type of home contact with sick people, who were possibly dependent for basic and instrumental activities of daily living, could have increased these ageist attitudes by making visible the possible disorder and lack of hygiene in the homes as a consequence of the decrease in care and assistance. Therefore, the professional demand of these people, due to the loneliness and dependence they suffer, which could add further stress to home care, should be emphasized.

Attitude improvement may be sufficient to increase positive attitudes, but negative ones may be more difficult to modify, necessitating meaningful learning experiences with older adults in multiple contexts to carefully plan how to interrupt negative perceptions that may emerge through the nursing education program [40].

### 4.2. Influencing Factors

Many articles have analyzed empathy, attitudes and knowledge [18,32,38], and have related various factors, such as sociodemographic parameters and exposure to older adults, as regulators of attitudes towards the elderly or the desire to work in the geriatric field, but the lack of longitudinal or evaluative research highlights a gap in the literature [34]. Our results indicate that clinical placement appears to be essential in changing attitudes, regardless of the factors that may interfere. Further work is needed to understand and evaluate the long-term effects and benefits of teaching strategies and activities used to improve the clinical practices of students in nursing homes [34].

Another aspect to highlight is that despite the increase in attitude, this did not change the choice of the geriatric area as a possible future job position. This aspect is still under study, and it seems that two factors influence the shortage of nurses who want to work in the geriatric field [41]. The first is the limited professional staff trained in the geriatric area, and the other is the limited training of students in all areas of work with older adults. While some studies observe that student interaction programs with older adults result in a positive increase in attitude towards working in long term care, other studies indicate that it also depends on the experience of the student and the skills and knowledge of the clinical instructor. The status of geriatric nursing and the current curricula—with emphasis on acute and critical care—reinforces student preference for these areas of work [42]. However, it is the older person who makes the greatest use of health services, whether in the hospital, in primary care, or in the long-term care setting. Although most students consider the geriatric setting as long term care, the increasing proportion of the older population worldwide, and its potential increased use of health care services, creates the need for age-appropriate curricula and specialized geriatric nurses to serve as mentors and role models, in addition to helping nursing students identify career advancement opportunities in gerontological nursing [42].

### 4.3. Limitations

The present study has a series of limitations, the first being the lack of a control group to determine the real effect of the intervention. The nature of the study did not allow for a control group, however. A second issue was the lack of analysis of the increase in knowledge with a specific tool. Aspects that could interfere with attitudes, such as the attitude of nurse educators and clinical instructors, were likewise not analyzed.

## 5. Conclusions

The attitude of the nursing students towards older adults was positive throughout their training, with an increase in positive attitudes during their studies, mainly after the integration of clinical practice with older adults, both in hospital and at nursing home level. Attitude improvement may be sufficient to increase positive attitudes, but negative ones may be more difficult to modify. Further studies are needed to understand all the factors involved and to design a curriculum destined to improve the negative attitude of nursing students towards older persons.

## Figures and Tables

**Figure 1 medicina-58-01201-f001:**
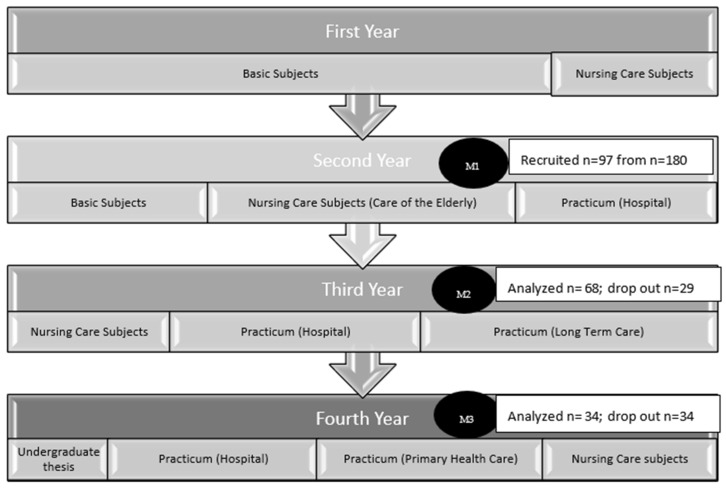
Flow chart of the study and curriculum of the nursing degree 2018.

**Figure 2 medicina-58-01201-f002:**
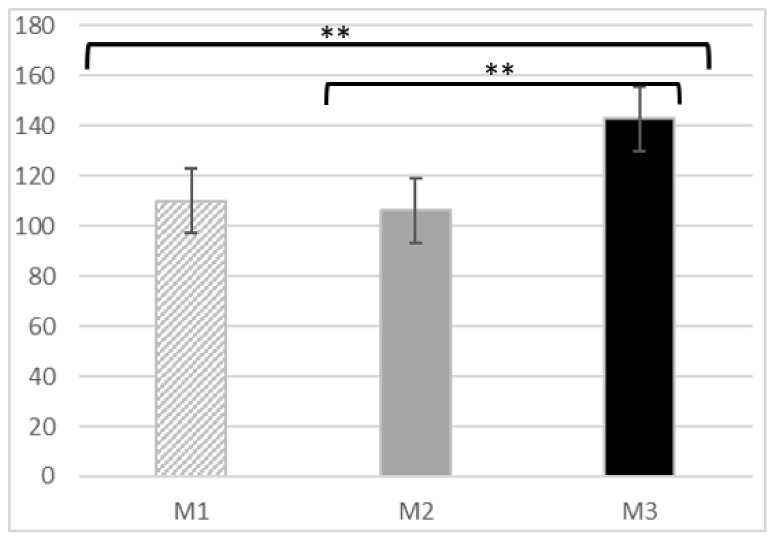
Kogan’s score throughout the study. M1: Moment 1; M2: Moment 2; M3: Moment 3; ** *p* < 0.001.

**Figure 3 medicina-58-01201-f003:**
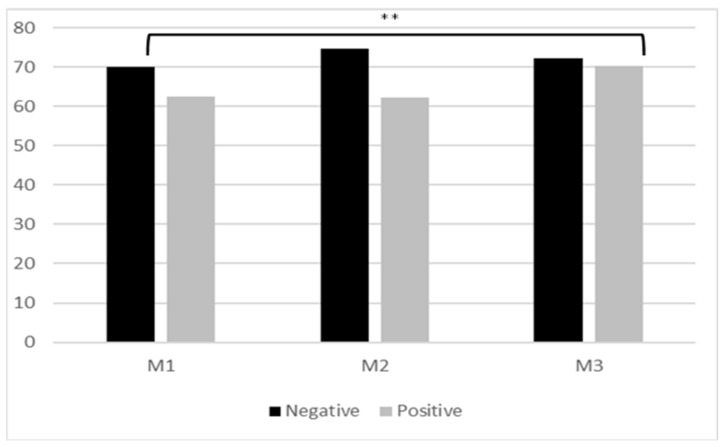
Kogan’s score in negative and positive subscales throughout the study. M1: Moment 1; M2: Moment 2; M3: Moment 3; ** *p* < 0.001.

**Figure 4 medicina-58-01201-f004:**
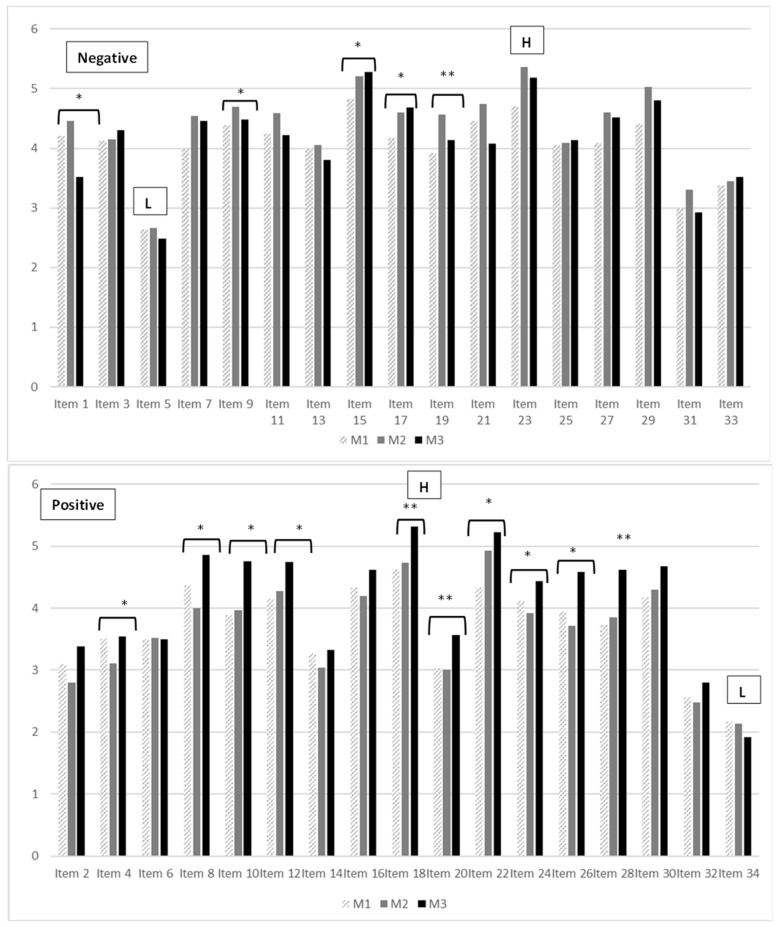
Negative and positive items score in Kogan’s scale throughout the study. H: highest scores, L: lowest scores; M1: Moment 1; M2: Moment 2; M3: Moment 3; * *p* < 0.05; ** *p* < 0.001.

**Figure 5 medicina-58-01201-f005:**
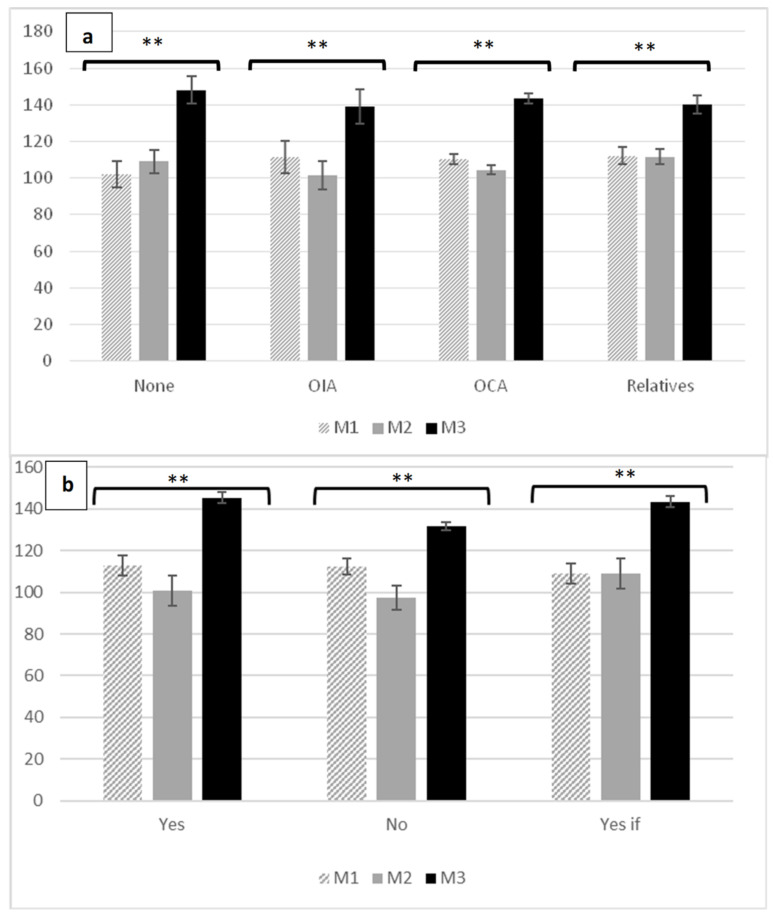
(**a**) (previous experiences): M1: Moment 1; M2: Moment 2; M3: Moment 3; OIA: previous experience with older institutionalized adults; OCA: previous experience with older community adults; Relatives: previous experience with older relatives at home; ** *p* < 0.001. (**b**) (preference of work): M1: Moment 1; M2: Moment 2; M3: Moment 3; Would you like to work with older people after your studies? Three possible answers: Yes, No, or Yes if: Only if I do not find work in another area; ** *p* < 0.001. Kogan’s score in the presence of previous experiences with older people and preference of work throughout the study.

**Table 1 medicina-58-01201-t001:** Characteristics of the study sample.

Variable	Mean (SD)/n (%)
Age, years	22.25 (4.92)
Sex	
Female	75 (77.3)
Male	22 (22.7)
Do you have previous experience caring for relatives or living with older people?	
None	10 (10.3)
Previous experience with OIA	12 (12.4)
Previous experience with OCA	56 (57.7)
Previous experience with older relatives at home	19 (19.6)
Would you like to work with older people after your studies?	
Yes	19 (19.6)
No	11 (11.3)
Only if I do not find work in another area	67 (69.1)

OCA: older community adults; OIA: older institutionalized adults.

## Data Availability

Data available on request from the authors.

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
