# Peer review of "Caring for the Elderly Enhances Positive Attitudes Better Than Knowledge in Nursing Students"

_medicina, 2022, doi:10.3390/medicina58091201_

Round 1

Reviewer 1 Report

Interesting research problem, but lack of essential elements, for example:

- lack of a control group,

- lack of analysis of the increase in knowledge with a specific tool,

- no variables that could affect the results: attitude of nurse educators and clinical instructors; 

which makes this research less valuable.

It is valuable that the authors are self-conscientious about it.

It is not necessary to present the research questions in detail - table 2.

Author Response

We would like to thank the reviewer for the critical review and valuable comments, which we have taken into account in this revised manuscript. Itemized responses are listed below. All the modifications have been marked with track changes throughout the manuscript to facilitate review.

Interesting research problem, but lack of essential elements, for example:

- lack of a control group,

- lack of analysis of the increase in knowledge with a specific tool,

- no variables that could affect the results: attitude of nurse educators and clinical instructors;

which makes this research less valuable.

It is valuable that the authors are self-conscientious about it.

Author´s answer: Thank you for your comment. The authors are conscientious about it, and we have included the above elements in the limitations section.

It is not necessary to present the research questions in detail - table 2.

Author´s answer: Thank you for your comment. The authors have deleted the table 2 and we have attached it as a supplementary file.

Reviewer 2 Report

In the abstract, you need to add one sentence about the background of the study, not only the aim. 

I could not find the gap in the study; can you please add the gap?

There is a need for further explanation of your study's recommendations (maybe you can add on conclusion section).

The originality check rate is about 35%, as attached.  

Author Response

REVIEWER 2

We would like to thank the reviewer for the critical review and valuable comments, which we have taken into account in this revised manuscript. Itemized responses are listed below. All the modifications have been marked with track changes throughout the manuscript to facilitate review.

In the abstract, you need to add one sentence about the background of the study, not only the aim. 

Author´s answer: Thank you for your comment. We agree with the reviewer, the authors have added the following in the abstract, lines 16 and 17: “There is a growing interest in the measurement of attitudes towards older people in healthcare professionals, as there is a need to implement interventions to improve ageist attitudes.”

I could not find the gap in the study; can you please add the gap?

Author´s answer: Thank you for your comment. We agree with the reviewer, the authors have reworded the last paragraph in the Introduction section: “It is convenient to know which interventions (taking theoretical subject and clinical placement in the geriatric setting) in the nursing curriculum can modify the attitude of nursing students in order to be able to plan the most appropriate interventions”.

There is a need for further explanation of your study's recommendations (maybe you can add on conclusion section).

 Author´s answer: Thank you for your comment. We agree with the reviewer, the authors have added the following in the Conclusions: “Attitude improvement may be sufficient to increase positive attitudes, but negative ones may be more difficult to modify. Further studies are needed to understand all the factors involved and to design a curriculum destined to improve the negative attitude of nursing students towards older persons.”

The originality check rate is about 35%, as attached.  

Author´s answer: Thank you for your comment. The authors have checked the manuscript, we have modified some paragraphs and deleted the Table 2 (Kogan Scale).